# Perceived Exertion during Moderate and Vigorous Physical Activity While Mask Wearing: A Quantitative and Qualitative Pilot Study

**DOI:** 10.3390/ijerph19095698

**Published:** 2022-05-07

**Authors:** Jennifer L. Scheid, Corinne Edwards, Michael Seils, Sarah L. West

**Affiliations:** 1Department of Health Promotion, Daemen University, Amherst, NY 14226, USA; 2Department of Physical Therapy, Daemen University, Amherst, NY 14226, USA; mseils@daemen.edu; 3Department of Athletic Training, Daemen University, Amherst, NY 14226, USA; corinne.edwards@daemen.edu; 4Kinesiology Program, Department of Biology & Trent/Fleming School of Nursing, Trent University, Peterborough, ON K9L 0G2, Canada; sarahwest@trentu.ca

**Keywords:** masks, facemask, pandemic, COVID-19, face covering, perceived exertion, physical activity

## Abstract

There is limited research examining the perception of exertion during exercise while wearing a facemask. The current study examined if mask usage during moderate or vigorous physical activity (MVPA) changed the self-reported perception of exertion. Seventy-two adults (18 years and older) who were physically active before the COVID-19 pandemic completed a questionnaire that assessed exercise habits and perceptions of mask wearing during MVPA. Participants reported their ratings of perceived exertion (RPE, on a scale of 1–10) while exercising. Wearing a mask resulted in higher RPE vs. no mask during both vigorous (8.4 ± 0.2 vs. 7.4 ± 0.1; *p* < 0.001) and moderate PA (6.6 ± 0.2 vs. 5.6 ± 0.2; *p* < 0.001). Qualitative analysis revealed mostly negative perceptions of exercising while wearing a mask, including respiratory issues, detriments to cardiovascular endurance, and general discomfort. A total of 40% of participants reported that they stopped exercising in an indoor/public setting because of a mask mandate in their region. Participants reported participating in less vigorous PA (4.7 ± 0.4 vs. 4.0 ± 0.4 h/week; *p* = 0.046), but not less moderate PA (3.3 ± 0.3 vs. 3.0 ± 0.3 h/week; *p* = 0.443) pre vs. during the pandemic. Our study suggests that facemask usage during MVPA causes an increase in RPE and may be one reason for a decrease in vigorous PA during the COVID-19 pandemic.

## 1. Introduction

On 11 March 2020, the World Health Organization declared COVID-19 to be a global pandemic. Many countries, including the United States and Canada, entered a lockdown. Over the course of the pandemic, modified reopening plans for public settings (such as gyms/fitness centers) were implemented. While the World Health Organization did not formally recommend that masks should be worn while exercising [1], the Centers for Disease Control and Prevention (CDC) stated that it may be difficult to wear a mask during high intensity physical activities [2]. Nevertheless, when gyms and fitness centers reopened in various capacities over the course of the COVID-19 pandemic, many geographic regions required that a facemask be worn while visiting and exercising at these establishments.

In individuals without disease that could impact the respiratory system, i.e., cardiopulmonary disease, there is a minimal physiological impact of wearing a facemask during moderate or vigorous physical activity (MVPA) [3,4,5]. For example, moderate PA (walking on a treadmill at 2.5 m/h) while wearing an N95 respirator, one of the most restrictive types of masks, did not appear to have any effect on ventilation in adults [6]. One study that examined military grade respirators suggested that there may be alterations to ventilation during high exercise intensities above 85% of maximum oxygen consumption [7]. More recently, less restrictive masks (i.e., cloth and surgical), the more commonly used masks by the public during the COVID-19 pandemic, were evaluated in heathy individuals during cycling to exhaustion. In this study of 14 adults, wearing a cloth or surgical face mask had no impact on exercise related physiological outcomes, including time to exhaustion, peak power, arterial oxygen saturation, or heart rate [3]. In general, the physiological effects of wearing a mask are minimal and are unlikely to impact the exercise performance in healthy individuals without cardiopulmonary disease [5]. 

Despite the minimal physiological impact of mask use during MVPA [5], anecdotally there is public resistance in some regions to wearing masks during exercise. However, there is limited research formally examining the perception of exertion during exercise while wearing a facemask, and these studies report mixed findings [3,8,9]. In one study, healthy individuals (*n* = 14) performed cycling to exhaustion under three conditions (surgical mask, cloth mask, no mask), and there was no difference in ratings of perceived exertion (RPE) based on mask wearing [3]. Similarly, healthy individuals (*n* = 16) performed a maximal cycling exercise test under three conditions (surgical mask, N95 respirator, no mask) and demonstrated no difference in RPE at any stage of the ramped protocol [8]. However, Wong et al. demonstrated that RPE was increased during a 6 min graded (10%) treadmill walking task between those wearing masks compared to those without masks [9]. Previous research has highlighted that during the COVID-19 pandemic, there may be some psychological barriers to mask use that may also impact exercise [10]. In the current study, we sought to examine the perception of exertion during exercise more thoroughly, using both quantitative and qualitative methods to examine mask perceptions. 

Therefore, the purpose of the current study was to use a mixed methods approach to examine if facemask use impacts the perception of exertion during MVPA. Quantitatively, we determined self-reported RPE with and without a facemask during both moderate and vigorous exercise. Qualitatively, we examined personal experiences with masks during exercise by asking participants about their experiences exercising while wearing a mask.

## 2. Materials and Methods

### 2.1. Participants

Participants were adults (over 18 years of age) who identified themselves as being regularly physically active before the COVID-19 pandemic (11 March 2020). Regular physical activity was defined as 150 min/week of MVPA for at least 6 months [11]. In addition to targeting participants who were physically active before the pandemic, we also used questions about physical activity to confirm that the participants met the ACSM’s guidelines for being physically activity before the COVID-19 pandemic [11]. We targeted people who exercised regularly because we expected them to have the most experience with exercising with a mask on and were therefore more appropriate to elicit perceptions of wearing masks during exercise (compared to individuals who were new to exercise). 

Participants were recruited via social media (Facebook and Instagram) and email using an electronic flyer which explained inclusion criteria (18 years of age and physically active for at least 6 months before the start of the pandemic, 11 March 2020). Recruitment flyers were also placed outside of the gym and fitness areas local to the Amherst NY region, and on a post-secondary college campus. Participants accessed the study using a link to the survey, which included collecting demographic, medical, and facemask usage information, in addition to requesting responses to physical activity questions using a modified version of the World Health Organization’s Global Physical Activity Questionnaire [12] and the Rate of Perceived Exertion (RPE) scale [11]. The survey was open from June 2021 through November 2021. The college’s institutional review board approved the study (Daemen University Human Subject Research Review Committee, ATH.0621.005, on 14 June 2021), and informed consent was obtained from all the subjects via the survey tool prior to study participation.

### 2.2. Data Collection

#### 2.2.1. Physical Activity

The survey provided to participants can be found in the Appendix A. The World Health Organization’s Global Physical Activity Questionnaire was modified to collect information about recreational moderate physical activity and recreational vigorous physical activity [12]. To collect information about physical activity prior to COVID-19, we stated the following before we asked specific questions about physical activity: “Please think about any vigorous exercise (an activity that causes large increase in breathing or heart rate) before COVID-19,” and “Please think about any moderate exercise (an activity that caused a small increase in breathing or heart rate) before COVID-19”. To collect physical activity information during COVID-19, we asked the same questions, but specified that we were inquiring about current physical activity habits.

#### 2.2.2. Rate of Perceived Exertion

We used a modified RPE scale of 1–10 to measure the participant’s perception of exertion when engaging in MVPA before the COVID-19 pandemic, and also their RPE when currently engaging in MVPA (i.e., during the COVID-19 pandemic) [11]. Furthermore, we also asked participants who wore a mask during MVPA to rate their RPE while wearing a mask.

#### 2.2.3. Personal Experiences with a Mask during Exercise

To learn more about the perceptions and personal experiences of participant’s mask wearing during exercise, our survey included two open ended-questions: (1) “If your workout routine changed because of a mask mandate, please elaborate about how your workout routine changed because of the mask mandate” and (2) “Please tell us about your experience with working out with a mask on”. Participants could enter their text into a provided textbox telling us about their experiences, as requested.

#### 2.2.4. Quantitative Data Analysis 

Only participants that reported at least 150 min of MVPA before the pandemic were included in the analysis. Demographic data was presented using descriptive and dispersion statistics. Repeated measures ANOVA was used to compare RPE scores (with and without a facemask, during both moderate and vigorous physical activity) and physical activity minutes (before COVID-19 and during the COVID-19 pandemic). Repeated measures ANOVA was used so that we could conduct the analysis while controlling for pre-existing medical conditions or type of mask worn during exercise. All data were analyzed using the SPSS for Windows (version 28.0, Chicago, IL, USA) statistical software package. Data are reported in mean ± SEM and significance was accepted at *p* < 0.05.

#### 2.2.5. Qualitative Data Analysis

Participants were provided with pseudonyms, and confidentiality was maintained with their individualized responses. The researcher reviewed and analyzed the responses for both open-ended questions: (1) “If your workout routine changed because of a mask mandate, please elaborate about how your workout routine changed because of the mask mandate” and (2) “Please tell us about your experience with working out with a mask on”.

The qualitative analysis focused on assessing themes for both open-ended questions regarding participant’s routine changes during the mask mandate and perceptions of exercising with masks on. The researcher read each line of the participant’s responses and identified key words and phrases. Open coding was used for initial data analysis to categorize responses. Axial coding was then used to narrow down codes into themes and subthemes [13]. A qualitative analysis of both open-ended questions was manually coded. Upon completion of the analysis, the qualitative findings, themes, and subthemes were organized and are displayed in Table 1 and Table 2.

One researcher was solely responsible for analyzing the data without the use of any external coding programs or intercoder assessment. The exploratory nature of the investigation alleviated concern for inter-rater reliability. No additional raters or coding programs were utilized in this study.

## 3. Results

### 3.1. Demographic Information

A total of 76 surveys were completed, 1 was excluded as it was a duplicate submission, and 3 were excluded because they did not meet the exercise requirements when they answered the exercise questions within the survey. Therefore, a total of 72 regular exercisers (prior to COVID-19) were included in the current analysis. A total of 98% of the participants resided in the United States. Participants primarily identified as female (73%), with a mean age of 33.5 ± 1.7 years (range 18–66 years), and were predominantly (91%) Caucasian. Twelve percent of the participants were current college athletes. Participants self-reported engaging in 3.3 ± 0.3 h/week of moderate PA and 4.7 ± 0.4 h/week of vigorous PA prior to the COVID-19 pandemic. While all participants engaged in regular physical activity (defined as at least 150 min of physical activity a week) as a study entry criterion, 25 of the participants reported a pre-existing medical condition with asthma (*n* = 15) as the most commonly reported condition. The other reported medical conditions were high cholesterol (*n* = 3), hypertension (*n* = 1), osteoporosis (*n* = 1), or “other” (*n* = 5). Three participants checked more than one medical condition. The participants reported wearing a variety of masks while being physically active during the pandemic. Of those who wore a mask during exercise, the most commonly reported mask types used during PA were cloth masks (41%), surgical masks (37%), and sport specific masks (11%).

### 3.2. Ratings of Perceived Exertion

Figure 1 shows that during vigorous PA, individuals indicated that their RPE (on scale of 1–10) while exercising and wearing a mask was higher compared to when they exercised with no mask (8.4 ± 0.2 vs. 7.4 ± 0.1; *p* < 0.001). During moderate PA, self-reported RPE while exercising and wearing a mask was also higher compared to when they exercised with no mask (6.6 ± 0.2 vs. 5.6 ± 0.2; *p* < 0.001). Results were still significantly different (*p* < 0.01), i.e., RPE was higher when wearing a mask compared to no mask during both vigorous and moderate exercise when controlling for pre-existing medical conditions or type of mask worn during exercise.

### 3.3. Physical Activity

A total of 41% of participants reported that they stopped exercising in an indoor/public setting because of a mask mandate in their region. Figure 2 demonstrates that individuals participated in less self-reported vigorous PA (4.7 ± 0.4 vs. 4.0 ± 0.4 h/week; *p* = 0.046) but not less moderate PA (3.3 ± 0.3 vs. 3.0 ± 0.3 h/week; *p* = 0.443) or total physical activity (8.0 ± 0.6 vs. 7.0 ± 0.6 h/week; *p* = 0.053) pre vs. during the pandemic. Prior to the COVID-19 pandemic, 100% of the participants met the ACSM recommendation for aerobic physical activity (this was a requirement to be included in the research study) and during the COVID-19 pandemic, 87.7% of the participants still met the ACSM recommendations for aerobic physical activity.

### 3.4. Personal Experiences with Mask Wearing 

The responses for perceptions of exercising while wearing a mask were analyzed into themes and subthemes, and outlined in Table 1. The overall perceptions of wearing masks were primarily negative. There were 2 individuals with positive experiences, 13 with neutral experiences, and majority of individuals that specifically addressed negative perceptions.

#### 3.4.1. Positive Experiences with Mask Wearing

Two individuals reported positive experiences associated with mask wearing while exercising, including increasing their strength and endurance while wearing the mask. Participant 2 reported, “I definitely believed it boosted my endurance levels by a lot”. Participant 50 echoed similar sentiments when describing feeling stronger compared to the pre-mask mandate. The participant attributed this feeling to a “placebo or it could be a hypoxic effect” due to mask wearing. However, these were the only two individuals that expressed positive experiences when describing mask wearing during exercise.

#### 3.4.2. Neutral Experiences with Mask Wearing

Thirteen individuals reported a neutral experience when describing mask wearing during exercise. These participants described the masks as tolerable and expressed that there were no changes to their routines. Participant 28 reports, “working out with a mask on was uncomfortable, but not enough to keep me from doing it”. Participant 74 described it as, “more of an inconvenience than effecting my workout.” In summary, 13 total individuals reported a generally neutral experience when describing their experiences of mask wearing during exercise.

#### 3.4.3. Negative Experiences with Mask Wearing

The most significant findings of this open-ended question were the negative perceptions of exercising while wearing a mask. These negative experiences were coded into categories of respiratory, discomfort, cardiovascular endurance, and decline in the desire to exercise while wearing a mask. 

Respiratory complaints were the most common theme observed. In total, 38 individuals reported negative experiences associated with their respiratory systems, including subthemes of difficulty breathing, discomfort, increased respiration rate, asthma symptoms, and the sensation of suffocating at times. Participants 5, 26, 32, 34, 42, and 49 all used the term “difficult” when describing their respiratory status while exercising with a mask. Participant 34 elaborated further, “sometimes it feels like its more difficult to exhale than inhale with a mask on”. Participants 35 and 60 specifically used the descriptor “discomfort” when describing their respiratory symptoms. 

As individuals described perceived changes to their respiratory rates. Participant 32 explains “When moving in general with a mask on, it increases my breathing rate a lot more than if I was not wearing a mask”. As breathing rates increased, individuals reported increased asthma-like symptoms and the sensation of suffocating. Participant 60 explained, “the repetitive exercise induced asthma attacks worsened”. Participant 69 explained that “running with a mask on is suffocating”.

Discomfort was the second most common theme of negative perception described. There were 27 participants that expressed discomfort, including sweat, increased temperature, light-headedness, and increased anxiety. Participant 5 described the “mask being soaked in sweat,” and Participant 16 reported, “the mask was irritating to wear”. The increased temperature was often described as an inconvenience and a challenge. Participants 33, 34, and 48 all described the mask as feeling hot and uncomfortable. Participants 23 and 63 described feeling light-headed while wearing masks while exercising, and Participant 8 explained that he/she “felt more anxious” due to the mask. These were common findings expressed by the 27 participants as negative perceptions of wearing masks while exercising.

Cardiovascular changes were the third most common negative perception of wearing masks while exercising. There were 21 participants that commented on the effects of cardiovascular endurance, including increased challenge, increased frequency of breaks needed for rest or hydration, increased fatigue, and decreased ability to work as hard. Participants 32, 49, and 61 all commented on requiring more breaks, attributed to wearing masks. Participant 51 reported that his/her “heart rate increased more rapidly”. Negative perceptions were described for numerous participants related to the effects on their cardiovascular system during exercise with masks on. 

Last, there were 5 participants that refused to exercise with masks on (Participants 37, 68, 70, 75, and 76). All these participants explained they felt so negatively about mask wearing during exercise that they refused to do it. Participant 75 reports, “never did it and don’t plan on trying it,” and participant 76 echoed similar sentiments when stating “I don’t do it”.

These themes and subthemes demonstrate the lived experience and perceptions of individuals concerning exercising with masks on.

### 3.5. Personal Experiences with Workout Changes Due to the Mask Mandate 

The responses from the participants for perceived workout changes due to the mask mandate were analyzed into themes and subthemes, as outlined in Table 2. The two themes found were: perceived workout changes due to the mask mandate, and a change in the location of exercise.

#### 3.5.1. No Changes in Exercise Routines 

There were 7 participants that reported no changes in their exercise routines (Participants 23, 33, 41, 53, 54, 56, 69). Participant 69 explains that he/she is “still running the same amount,” and participant 33 says, “I still lifted inside at the gym”. These 7 individuals did not express any workout changes due to the mask mandate.

#### 3.5.2. Negative Changes in Exercise Routines 

There were 19 individuals who expressed negative changes to workout frequency, intensity, and duration. Participant 12 explains, “frequency of indoor exercise and exercise intensity decreased”. Participant 16 specifically reported the need to “stop HIIT workouts after my lifting session because wearing a mask would make it hard to breath.” All these individuals expressed different scenarios of strategically decreasing their frequency, intensity, or duration of exercise due to the mask mandates. 

#### 3.5.3. Location Change of Exercise 

One of the most common themes identified from data analysis was a change in location for exercise. Of the participants, 14 described increased exercising at home, and 11 described increased exercising outside. Many individuals reported quitting the gym and focusing solely on exercise at home. These participants purchased equipment and were able to establish and build an exercise routine in the convenience of their homes. The most common exercises reported at home included using cycling, treadmill/elliptical equipment, and body weight supported exercise. Participant 36 explains, “I did a lot more working out at home. Using the stationary bike and treadmill more, increasing body weight exercises and adding in yoga”. Participant 68 describes, “I bought home gym equipment (weights, spin bike, elliptical) and workout at home”.

There were 11 individuals that emphasized increasing their outdoor walking and running routine. For example, Participant 76 reports, “I now use a home gym exclusively, or fitness walk outside”. Many individuals reported they avoided indoor activity and focused on outdoor work instead. Participant 62 says that he/she “quit the gym and worked out at home and continue outdoor running”.

## 4. Discussion

The purpose of the current study was to use a mixed methods analysis of a survey to examine if facemask use due to COVID-19-related mandates impacts the perception of exertion during MVPA. Quantitatively, we examined RPE with and without a facemask and demonstrated that RPE was higher while wearing a mask during both moderate and vigorous PA. Qualitative analysis provided further insights and revealed that participants had mostly negative perceptions of exercising while wearing a mask. Negative perceptions included a perceived increase in respiratory issues, detriments to cardiovascular endurance, and general discomfort due to the mask during exercise.

This is the first mixed methods study to explore mask usage during the COVID-19 pandemic and the associated perceptions of exertion during PA. Our results of increased RPE with exercise and mask usage are not surprising, given that masks have been reported to be uncomfortable, even in non-exercise related situations (for example, for healthcare workers). Headaches [14], acne [15], nasal bridge scaring [16], facial itching [16], rash/irritation [17], and discomfort related to increased facial temperatures [18,19] have been reported due to mask wearing. Previously, few studies have examined RPE during exercise, and existing studies reported varying results [3,8,9]. For example, Shaw et al. prospectively examined RPE in healthy individuals during cycling to exhaustion while wearing a mask, demonstrating that relative to peak performance, there was no difference in RPE based on mask wearing (surgical mask, cloth mask, or no mask) [3]. Similarly, Epstein et al. also demonstrated that in healthy individuals (*n* = 16), there were no differences in RPE when the participant performed a maximal cycling exercise test under three conditions (surgical mask, N95 respirator, or no mask) [8]. However, Wong et al. demonstrated that RPE was increased during a 6 min graded (10%) treadmill walking task between those wearing masks compared to those without masks [9]. These differences regarding the impact of mask use on RPE during PA may have to do with the subjective nature of RPE scales. 

While RPE scores are subjective, they do evaluate the perception of exhaustion, which may be related to physiological changes such as elevations in heart rate, but they may also vary based on personal experiences. Not only did our study examine RPE changes, but we also explored the perceptions of the individual participants while exercising with a mask. The participants in the current study reported more negative statements (91 statements) regarding exercising with masks on in comparison to positive (2 statements) or neutral perceptions (13 statements). The most common negative themes include the affects associated with the respiratory system, followed by discomfort and a decline in cardiovascular endurance. There were five individuals who expressed such negative feelings for exercising with masks on that they adamantly refused to exercise while wearing masks. 

Perhaps not surprisingly, individuals reported participating in less vigorous PA during the COVID-19 pandemic compared to vigorous PA prior to the COVID-19 pandemic. Following the declaration of the pandemic, PA levels in the general active adult population have reportedly decreased [20,21], and there was a steep decline in step counts worldwide [22]. Interestingly, in the current study, while vigorous PA decreased pre vs. during the COVID-19 pandemic, there were no reported changes in moderate or total PA. It is important to note that the participants in the current study all participated in physical activity (meeting ACSM guidelines) before the COVID-19 pandemic and potentially had enough experience with exercise to continue their exercise routine during the COVID-19 pandemic, but were unable to continue with the same quantity of vigorous PA. The qualitative analysis in the current study also suggests that the participants changed their workout routines to include more exercise at home or outdoors. It is possible that this shift in the location of exercise (and potentially, the mode of exercise) contributed to the change in exercise intensity. Cortis et al. recently examined differences between a Zumba class (60 min class at a gym with an instructor) vs. a Zumba Exergame session (also 60 min) and demonstrated that the exercise heart rate was higher during the in-person Zumba class, suggesting that the increased exercise intensity during the class compared to the game could be due to the real-time feedback and the encouragement and enthusiasm of the instructor and other participants [23]. In the current study, more at-home workouts may have resulted in less vigorous exercise if the individual was used to feedback and encouragement from instructors, trainers, or peers in a gym setting.

In addition to reporting that some participants changed their workout routine to include more exercise at home or outdoors, some participants explicitly stated that they quit their gym memberships and changed their routine to include exercise at home and outdoors. There were some individuals in the current study that reported no changes to their exercise routines due to the mask mandates. However, more individuals expressed changing their routine to decrease exercise frequency, intensity, and duration of exercise due to the mask mandate. While many of the individual experiences varied, it is possible that these changes in exercise routine could negatively impact a participant’s physical activity and, more specifically in this study, lead to a reduction in the participant’s quantity of vigorous PA.

Our study includes a unique cohort of adults who regularly engaged in PA prior to the COVID-19 pandemic: 100% of the participants met the ACSM recommendation for aerobic PA. However, during the COVID-19 pandemic, 87.7% of the participants met the ACSM recommendations for aerobic PA. This indicates that even adults who are consistently physically active (i.e., are typically meeting the ACSM guidelines) reported challenges and barriers to exercising in a mask that may have then negatively impacted their ability to maintain their exercise levels. This barrier may, in fact, be amplified in individuals who were not exercisers prior the pandemic and would be looking to start their physical activity journeys; this is an important additional future research question to assess. Importantly, just under half (41%) of participants reported that they stopped exercising in an indoor/public setting because of a mask mandate in their region. Therefore, we observed that mask mandates do have an impact on the decision to participate in physical activity at a traditional gym or indoor fitness facility, which may influence the intensity of physical activity during exercise [23]. 

One limitation of this study is that it was not a prospective laboratory study, but rather a retrospective study where participants had to recall both their RPE and their physical activity levels before the COVID-19 pandemic and during the mask mandate. However, this study used the RPE information as only one outcome, and more thoroughly examined the perceptions of wearing a mask during exercise via the open-ended questions and associated qualitative analysis. From a public policy perspective, these opinions are important and impact individuals’ physical activity behaviors. Moreover, the current study assessed physical activity using questions from the modified World Health Organization’s Global Physical Activity Questionnaire [12]. We focused only on purposeful recreational physical activity and the Global Physical Activity Questionnaire (when completed in its entirety) assesses all aspect of physical activity, including physical activity involved in travel and work. These other aspects of PA were likely impacted by COVID-19 because of changes in office and work environments, not largely as a result of mask mandates. However, since this study only examined recreational activity, it does limit the generalizability of the research. Furthermore, the target population in this study was previously physically active because we expected this group to have the most experience with exercising with a mask on and they were, therefore, an appropriate group to elicit perceptions of wearing masks during exercise. However, these results may differ in individuals that are sedentary and who are just attempting to begin participating in an exercise program. Lastly, it is important to note that this data was collected during the COVID-19 pandemic, and evaluating perceptions during this time period may have caused increased variability compared to perceptions during non-pandemic times. Xie et al. demonstrated that during the COVID-19 pandemic, working memory capacity impacts an individual’s ability to comply with social distancing mandates [24]. Future research should evaluate if cognitive factors mediate the perceptions of face masks use during exercise. 

## 5. Conclusions

Our study suggests that facemask usage during MVPA causes an increase in RPE. Moreover, negative perceptions of wearing a mask during exercise are common, and are associated with perceptions of respiratory challenges, discomfort, and limitations of cardiovascular endurance. Therefore, negative perceptions of wearing a mask during exercise may be one factor contributing to a decrease in vigorous PA during the COVID-19 pandemic. The current study also suggests that individuals who regularly participated in MVPA prior to the COVID-19 pandemic changed their exercise routines during the facemask mandate. Future research should focus on how to reduce the perception of discomfort while wearing a facemask during exercise, since perceptions of these discomforts may have a significant negative impact on MVPA.

## Figures and Tables

**Figure 1 ijerph-19-05698-f001:**
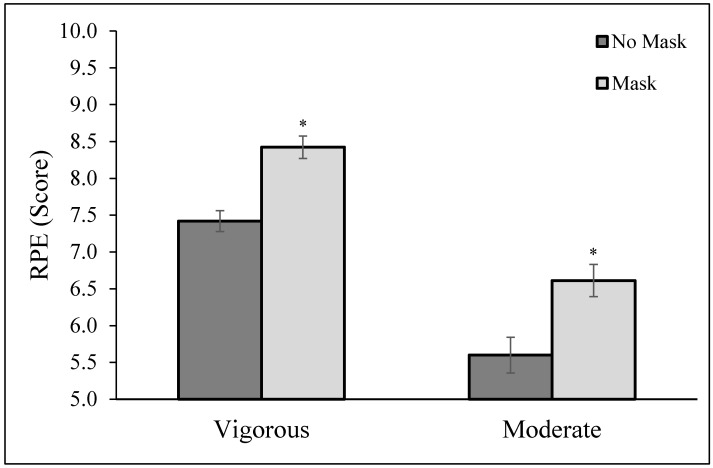
Ratings of Perceived Extension (RPE, scored on a scale of 1–10) during moderate and vigorous exercise with and without a mask. * *p* < 0.001 when comparing exercising with no mask on to exercising with a mask on.

**Figure 2 ijerph-19-05698-f002:**
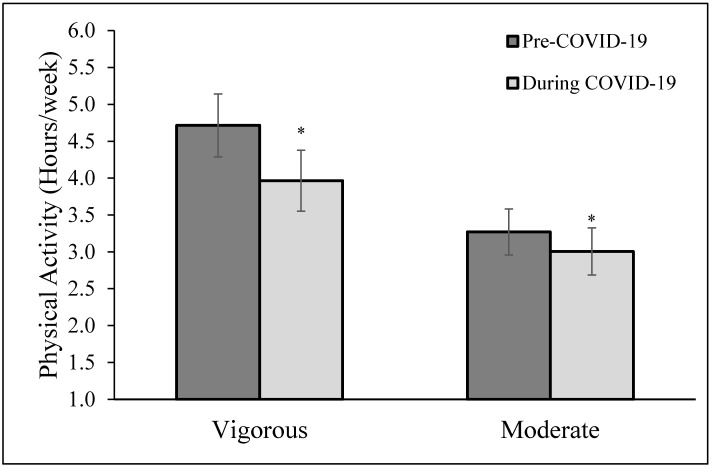
Physical activity (moderate and vigorous) before and during COVID-19. * *p* < 0.05 when comparing pre-COVID-19 pandemic self-reported exercise with during the COVID-19 pandemic self-reported exercise.

**Table 1 ijerph-19-05698-t001:** Overview of Common Themes for Perceptions of Exercising While Wearing a Mask.

Perception	Theme	Subtheme
Positive	Improvements (*n* = 2)	Increased Endurance Increased Strength
Neutral	No Change (*n* = 13)	No Changes to Routine Tolerable Small Learning Curve but Does not Impede Workout
Negative	Respiratory (*n* = 38)	Difficult to Breathe Respiratory Discomfort Increased Respiratory Rate Asthma Symptoms Increased Suffocating at Times
	Cardiovascular Endurance (*n* = 21)	Increased Challenge More Rest Breaks Needed Increased Hydration Needed Fatigues Faster Unable to Work as Hard
	Discomfort (*n* = 27)	Sweaty Mask Increased Temperature of Air Mask is Uncomfortable Increased Lightheadedness Increased Anxiety
	Declines to Exercise with Mask (*n* = 5)	Declines Exercising with Mask on No Plans to Exercise with Mask on

**Table 2 ijerph-19-05698-t002:** Overview of Perceived Workout Changes due to the Mask Mandate.

Perceived Changes	Theme	Subtheme
Exercise	No Change (*n* = 7)	No Changes Noted
	Decreased (*n* = 19)	Intensity Frequency Duration
Location	Increased Exercise at Home (*n* = 14)	Used Equipment at Home Purchased Equipment for Home
	Increased Exercise Outside (*n* = 11)	Increased Walking Outside Increased Running Outside

## Data Availability

The data presented in this study are available on request from the corresponding author.

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
