# Peer review of "Perceived Exertion during Moderate and Vigorous Physical Activity While Mask Wearing: A Quantitative and Qualitative Pilot Study"

_ijerph, 2022, doi:10.3390/ijerph19095698_

Round 1

Reviewer 1 Report

The article presents  a novel theme in the context of the pandemic in which we find ourselves. In addition, it is consistent in its elaboration. Perhaps it can be improved, clarified and deepened in two sections:

  • Theoretical support of the research
  • Qualitative methodology procedures
  •  

It is understandable that due to the theme of the article, the scientific references that support the work are more contextual than theoretical. Even so, an exhaustive search of all the existing research on the subject "use of a mask during physical activity" is not carried out.

The logic of qualitative data analysis is not made explicit in the work. This does not allow visibility to the work carried out. It would be advisable to show the processes followed for the construction of the categories of analysis.

Author Response

Reviewer 1

The article presents a novel theme in the context of the pandemic in which we find ourselves. In addition, it is consistent in its elaboration. Perhaps it can be improved, clarified and deepened in two sections:

Theoretical support of the research

Qualitative methodology procedures

It is understandable that due to the theme of the article, the scientific references that support the work are more contextual than theoretical. Even so, an exhaustive search of all the existing research on the subject "use of a mask during physical activity" is not carried out.

Response: An exhaustive search of the literature is beyond the scope of this primary research article.  However, we do highlight additional manuscripts that explore mask use during exercise in the discussion section of our paper below:

“This is the first mixed-method study to explore mask usage during the COVID-19 pandemic and associated perceptions of exertion during PA. Our results of increased RPE with exercise and mask usage are not surprising given that masks have been reported to be uncomfortable even in non-exercise related situations (for example in healthcare workers). Headaches [13], acne [14], nasal bridge scaring [15], facial itching [15], rash/irritation [16] and discomfort related to increased facial temperatures [17,18] have been reported while wearing a mask. Previously, few studies have examined RPE during exercise, and reported varying results [3,8,9]. For example, Shaw et al. prospectively examined RPE in healthy individuals during cycling to exhaustion while wearing a mask, and demonstrated that relative to peak performance, there was no difference in RPE based on mask wearing (surgical mask, cloth mask, vs. no mask) [3].  Similarly, Epstein et al. also demonstrated in healthy individuals (n=16) that there were no differences in RPE when participated performed a maximal cycling exercise test under three conditions (surgical mask, N95 respirator, vs. no mask). However, Wong et al. demonstrated that RPE was increased during a 6-min graded (10%) treadmill walking task between those wearing masks compared to those without masks [9]. These differences of the impact of mask use on RPE during PA may have to do with the subjective nature of RPE scales.”

The logic of qualitative data analysis is not made explicit in the work. This does not allow visibility to the work carried out. It would be advisable to show the processes followed for the construction of the categories of analysis.

Response: We agree with the reviewer that the methods needed more detail. We added the full survey that was giving to the participants as an appendix as an end of the manuscript. Additionally, we added more information to the Qualitative Data Analysis (section qualitative 2.2.4):

“The qualitative analysis focused on assessing themes for both open-ended questions regarding participant’s routine changes during the mask mandate and perceptions of exercising with masks on. The researcher read each line of the participant’s responses and identified key words and phrases. Open coding was used for initial data analysis to categorize responses. Axial coding was then used to narrow down codes into themes and subthemes [13]. A qualitative analysis of both open-ended questions was manually coded. Upon completion of the analysis, the qualitative findings, themes, and sub-themes were organized and are displayed in Table 1 and Table 2.”

Reviewer 2 Report

This is both an interesting and timely investigation regarding the novel impact of wearing a facemask during moderate and vigorous physical activity. The manuscript is well presented and communicated simplistically; it was easy to read and follow. The Method and Results were clear, along with the key findings in the Discussion. I have just one consideration regarding feedback on the manuscript:

  • The second paragraph in the Introduction refers to "In healthy individuals"; what constitutes an individual being deemed healthy? I believe an explanation of this term, or a description of what it represents would be beneficial.

Author Response

Reviewer 2

This is both an interesting and timely investigation regarding the novel impact of wearing a facemask during moderate and vigorous physical activity. The manuscript is well presented and communicated simplistically; it was easy to read and follow. The Method and Results were clear, along with the key findings in the Discussion. I have just one consideration regarding feedback on the manuscript:

The second paragraph in the Introduction refers to "In healthy individuals"; what constitutes an individual being deemed healthy? I believe an explanation of this term, or a description of what it represents would be beneficial.

Response: Thank you for the comment.  In this context we defined healthy individuals as individuals without disease.  We have made this clearer in the introduction, and the sentence now reads:

“In individuals without disease that could impact the respiratory system, i.e. cardiopulmonary disease, there is a minimal physiological impact of wearing a facemask during moderate or vigorous physical activity (MVPA) [3–5].”

Reviewer 3 Report

  • Very interesting, pertinent and current theme.
  • The mixed methodology is also to value- A Quantitative and Qualitative Pilot Study.
  • Summary: I think the first sentence of the summary is very questionable: "In healthy individuals the physiological impact of wearing a facemask during moderate or vigorous physical activity (MVPA) is minimal." This is because what they evaluated was the perception of individuals and not the physiological impact.
  • Being perceived Exertion during Physical Activity (Moderate and Vigorous) it is always limiting to be evaluating perceptions during a period of so much uncertainty, it would be interesting to promote some physical evaluation.
  • On line 67 there is a crow "... we reviewed the psychological and psychological impact of wearing a mask..." that it is necessary to review.
  • In materials and methods- Participant is presented as was defined a Regular physical activity "was defined as 150 min/week of MVPA for at least 6 months" but it is not defined as Vigorous physical activity.
  • The characteristics of the sample are not presented, for example, the age range, the type of PA, the percentage of individuals who practice Moderate and Vigorous PA… There is a high discrepancy in the gender of the participants (Participants primarily - 73% identified as female), a statistical evaluation should be carried out to ensure this heterogeneity.
  • The instruments used in the collection of information are not clearly presented.
  • In the results, demographic information, lines 163 and 164, "34.2% of the participants reported a pre-existing medical condition with asthma (19.2%) as the most commonly reported condition." The conditions are unclear.
  • And why aren't these variables analysed? 34% is a very high percentage. I think it might be very interesting to do so.
  • In line 397, Conclusions – "While wearing a facemask has not been found to lead to physiological changes in response to exercise in healthy adults," the initial comment reiterates.
  • In line 408- How can current physiological impacts be stated? I don't think that's possible.
  • Effectively one limitation of this study is that it was not a prospective laboratory study, rather a retrospective study, so talking about physiological issues does not seem appropriate to me, but rather perceptions. We are talking about memory warehouse activations in pre-world and pandemic times, with attention and memory issues easily influenced by uncertainty. The text should soon be adapted in this direction.

Author Response

Reviewer 3

The mixed methodology is also to value- A Quantitative and Qualitative Pilot Study.

Summary: I think the first sentence of the summary is very questionable: "In healthy individuals the physiological impact of wearing a facemask during moderate or vigorous physical activity (MVPA) is minimal." This is because what they evaluated was the perception of individuals and not the physiological impact.

Response: The reviewer is correct.  The sentence has been updated to reflect the focus on perceived exertion in this study.  The first sentence in the abstract now reads:

“There is limited research examining the perception of exertion during exercise while wearing a facemask.”

Being perceived Exertion during Physical Activity (Moderate and Vigorous) it is always limiting to be evaluating perceptions during a period of so much uncertainty, it would be interesting to promote some physical evaluation.

Response: We think this point is worth noting in the limitation section of the discussion.  The following was added to the last paragraph of the discussion:

“Lastly, it is important to note that this data was collected during the COVID-19 pandemic and evaluating perceptions during this time-period may have caused increased variability compared to non-pandemic times. Xie et al. [23] demonstrated that during the COVID-19 pandemic working memory capacity impacts an individual’s ability to comply with social distancing mandates.  Future research should evaluate if cognitive factors mediate the perceptions of face masks use during exercise.”

On line 67 there is a crow "... we reviewed the psychological and psychological impact of wearing a mask..." that it is necessary to review.

Response: We thank the reviewer to pointing us to this sentence.  We did not mean to imply that we were going to review the literature, rather we were citing a prior review article.  We updated this sentence to me more clear: “Previous research has highlighted that during the COVID-19 pandemic there may be some psychological barriers to mask use that may also impact exercise [10].”

In materials and methods- Participant is presented as was defined a Regular physical activity "was defined as 150 min/week of MVPA for at least 6 months" but it is not defined as Vigorous physical activity.

Response: Our goal was to recruit participants that exercise at least 150 min/week (either moderate or vigorous exercise).  MVPA is defined as moderate and/or vigorous exercise (second paragraph of the Introduction).

The characteristics of the sample are not presented, for example, the age range, the type of PA, the percentage of individuals who practice Moderate and Vigorous PA… There is a high discrepancy in the gender of the participants (Participants primarily - 73% identified as female), a statistical evaluation should be carried out to ensure this heterogeneity.

Response: We collected country of residence, age, race and college athlete status. We updated are demographic paragraph of the results to include the range and the percentage of current college athletes in the study:

“Ninety-eight percent of the participants resided in the United States. Participants primarily (73%) identified as female, with a mean age of 33.5±1.7 years (range 18-66 years) and were predominantly (91%) Caucasian. Twelve percent of the participants are current college athletes. Participants self-reported engaging in 3.3±0.3 hours/week of moderate PA and 4.7±0.4 hours/week of vigorous PA prior to the COVID-19 pandemic.”

The instruments used in the collection of information are not clearly presented.

Response: We agree with the reviewer that the methods needed more detail. We added the full survey that was giving to the participants as an appendix at the end of the manuscript.

In the results, demographic information, lines 163 and 164, "34.2% of the participants reported a pre-existing medical condition with asthma (19.2%) as the most commonly reported condition." The conditions are unclear.

Response:  We clarified the medical conditions in the results section.  The section is updated as:

“While all participants engaged in regular physical activity (defined as at least 150 min of physical activity a week) as a study entry criterion, 25 of the participants reported a pre-existing medical condition with asthma (n=15) as the most commonly reported condition. The other reported medical conditions were high cholesterol (n=3), hypertension (n=1), osteoporosis (n=1), or “other” (n=5). Three participants checked more than one medical condition.”

And why aren't these variables analyzed? 34% is a very high percentage. I think it might be very interesting to do so.

Response: We agree that these medical conditions could impact our finding, especially the RPE during exercise.  We repeated the analyses controlling for pre-exciting medical condition and did not find any differences in the results.  The following is included in the results (Section 3.2):

“Results were still significantly different (p <0.01), i.e., RPE was higher when wearing a mask compared to no mask, during both vigorous and moderate exercise when controlling for pre-existing medical conditions or type of mask worn during exercise.”

In line 397, Conclusions – "While wearing a facemask has not been found to lead to physiological changes in response to exercise in healthy adults," the initial comment reiterates.

Response: We agree with the reviewer. This sentence has been deleted.

In line 408- How can current physiological impacts be stated? I don't think that's possible.

Response: This statement was updated to only include perceptions, not physiological impacts:

Future research should a focus on how to reduce the perception of discomfort while wearing a facemask during exercise, since perceptions of these discomforts may have a significant negative impact on MVPA.”

Effectively one limitation of this study is that it was not a prospective laboratory study, rather a retrospective study, so talking about physiological issues does not seem appropriate to me, but rather perceptions. We are talking about memory warehouse activations in pre-world and pandemic times, with attention and memory issues easily influenced by uncertainty. The text should soon be adapted in this direction.

Response: We agree the findings in the current study only reflect perceptions of exertion and mask wearing. We deleted all information about physiology in the abstract, discussion, and conclusions.